# Enhancing the Fatigue Design of Mechanical Systems Such as Refrigerator to Reserve Food in Agroindustry for the Circular Economy

Seongwoo Woo [1,*], Dennis L. O'Neal [2], Yimer Mohammed Hassen [1] and Gezae Mebrahtu [1]

[1] Manufacturing Technology, Mechanical Technology Faculty, Ethiopian Technical University, Addis Ababa P.O. Box 190310, Ethiopia

[2] Department of Mechanical Engineering, School of Engineering and Computer Science, Baylor University, Waco, TX 76798-7356, USA

\* Correspondence: twinwoo@yahoo.com; Tel.: +251-90-047-6711

**Abstract:** To prolong the fatigue life of a product handled by machines such as refrigerators and agricultural machinery, parametric accelerated life testing (ALT) is recommended as a systemized approach to detect design inadequacies and reduce fatigue. It demands (1) an ALT strategy, (2) a fatigue type, (3) parametric ALTs with change, and (4) an estimate of whether the present product completes the BX lifetime. The utilization of a quantum-transported life-stress type and a sample size are advocated. The enhancements in the lifetime of a refrigerator ice-maker, containing an auger motor with bearings, were employed as a case study. In the 1st ALT, a steel rolling bearing cracked due to repeated loading under cold conditions (below $-20\,^\circ$C) in the freezer compartment. The bearing material was changed from an AISI 52100 Alloy Steel with 1.30–1.60% chromium to a lubricated sliding bearing with sintered and hardened steel (FLC 4608-110HT) because of its high fatigue strength at lower temperatures. In the 2nd ALT, a helix made of polycarbonates (PCs) fractured. In the redesign, a reinforced rib of the helix was thickened. Because no troubles in the 3rd ALT happened, the life of an ice-maker was proven to have a B1 life 10 years.

**Keywords:** mechanical product; fatigue design; parametric ALT; ice-maker; bearing; design flaws

## 1. Introduction

To have mechanical systems, such as refrigerators, be competitive in the marketplace, their designs should incorporate the newest scientific and engineering knowledge and perform to the satisfaction of customers in the marketplace. If a system or product with new features is introduced into the market with insufficient testing, there is the possibility for early failure of the product due to the inferior design of the new features in the product. These premature failures in the market will undesirably influence the perceived quality of the product and affect the nature and environmental change in the global market related to sustainability. Moreover, many consumers will be forced to discard the faulty product and produce useless garbage. In an ideal case, design defects would be identified before the product was introduced in the market. This identification process would help reduce waste because products would be used for their fully anticipated lifetime before being replaced. To avoid discovering unanticipated design defects after the product is released and reinforce the proposition of "zero waste" in the market, any new traits for a product should be evaluated in the development process before being launched to the end-user. Assessing the reliability of a new mechanical system should include a structured method with reliability quantitative (RQ) statements [1]. Improving the reliability of a product should reduce or prevent waste from poorly designed products. Thus, this procedure should help improve sustainability with regard to the launch of new products in the market.

One feature of many new refrigerators is the ice-maker. Typically, these include an auger motor with a bearing designed to attain sufficient torque to sufficiently crush ice for discharge through the dispenser. The auger achieves decreasing speed through several gears engaged with a driving gear mounted in the shaft. The auger must operate under the low temperature condition ($-20\ ^\circ$C↓) in the freezer compartment. It is subjected to repeated stresses supported by bearings. A common material utilized in the ball bearing rings is the alloy steel AISI 52100 because it is simply forged, heat-treated, and machined. This steel is commonly used in numerous products because of its durability [2–5] and, under normal circumstances, it can be integrated into designs so a product will provide a long life under normal circumstances for customers.

A lack of reliability can have severe negative consequences. For example, two Boeing 737 MAXs crashed, resulting in the death of 346 passengers. The plane was grounded from March 2019 to December 2020. The airplane used the CFM International LEAP-1B engines, adopting the most effective 68-inch fan design. They were 12% more fuel efficient and 7% more lightweight than previous engines [6]. Inspectors had conjectured that the accident was caused by the engine in the aircraft. As a result, the whole economy experienced the elimination of loss of parts (or wastes) due to an improper design. Possible troublesome components thus needed to be confirmed by laboratory testing to produce a reliability quantitative (RQ) expression [7–9].

Fatigue is the main origin of metallic failure in the structure, explaining approximately 80–95% of all failures [10]. It displays itself in the form of cracks which usually start from stress raisers, such as holes, slender surfaces, grooves, etc., on the structure of systems. Fatigue is the failure of a material that is frequently subjected to cyclic loading. Of particular concern is the failure of components during low-cycle fatigue, specifically in the area of turbine–engine that is composed of nickel-built polycrystalline matter [11,12]. It is also measured as a quantity element, such as the stress proportion, $R$ (=$\sigma_{min}/\sigma_{max}$), explained as the correlation of the greatest cyclic stress to the least cyclic stress [13]. Utilizing a stress proportion, $R$, presented in an ALT, should help identify the design defects in the mechanical system.

Designers have frequently recognized design imperfections and have fixed them by utilizing techniques such as Taguchi's method [14]. In particular, the design of experiments (DOE) [15] is an organized method used to discover the connection between the factors governing a procedure and its production. The goal is to confirm that the factors are placed in the most successful manner for performing (or environmental) situations. The DOE is executed for related factors that determine product designs. Their functionality is revealed by an analysis of variance (ANOVA). Because a person who operates a DOE may not know which factors are the most influential in a failure, there is no specified process for identifying fatigue failure in the calculations. Thus, the DOE may require a large number of mathematical calculations and may not identify a potential source of failure.

Designers have frequently utilized the strength of materials as a solution to help in a conventional design [16–18]. A crucial element in fracture mechanics [19] is toughness as a material attribute of strength. With the implementation of quantum mechanics, engineers have pinpointed that structural failures occur from nanoscale or microscale voids, which may occur in metallic alloys or engineering plastics. As finite samples and limited testing periods are utilized [20–24], this method cannot reproduce the design flaws in a complex form or identify the fatigue problems that may occur with the product because of the specific way consumers use the product in the market. To discover the fatigue phenomena in a system functioning by machinery, a life-stress type [25,26] can be integrated with a (quantum) mechanics way to distinguish a prevailing defect or crack form in matter because unsuccessfulness stochastically happens in the region of particularly big stress.

The finite element method (FEM) [27] is utilized in a different way. Designers propose that failures may be determined by (1) an appropriate mathematical (Lagrangian or Newtonian) formulation; (2) deriving the time response for loads, generating the stress/strain on the part structure; (3) employing the generally accepted method of rain-flow counts

with von Mises stress [28]; and (4) evaluating system effectiveness by Palmgren–Miner's principle [29]. Deploying this methodology shall give closed-formation answers. However, this method cannot pinpoint fatigue failures in a complex system produced by structural defects such as microvoids, sharp edges, slender surfaces, etc.

This investigation proposes parametric ALT as a straightforward approach to identify the structural flaws of a new product and improve the reliability of the product. It involves the following: (1) an ALT scheme developed on the BX lifetime, (2) load study, (3) ALTs with the structure alterations, and (4) an appraisal of whether the system structure fulfills the objective BX lifetime. This procedure for recognizing the root causes and enhancing design examines properness in the mechanical products such as refrigerators, agricultural machinery, automobiles, etc. By eliminating problematic materials (AISI 52100 Alloy Steel) from being used in a product, it can help improve the reliability of the product, and reduce waste because the consumer is able to utilize the product longer. Furthermore, this experimental, computational, and theoretical research relating to natural and applied engineering will improve the human lifestyle in economics, social sciences, and the humanities and allow forecasts and effect evaluations of worldwide transform and evolution related to sustainability. The quantum-transported failure type and sample size are also advocated. A new refrigerator ice-maker involving an auger motor with a bearing is used as a case study.

## 2. Parametric ALT for a Product Functioned by Machine

### 2.1. Meaning of BX Lifetime

A product operated by machinery utilizes generated power to achieve a desired motion by adapting an appropriate mechanism [30]. Forces are utilized to supply the movement of mechanisms in the system. This movement signifies that the system shall be subjected to repeated loading. In a mechanical product, fatigue failure occurs when there are structural imperfections such as notches, sharp edges, grooves, and slender surfaces in a component.

For instance, a refrigerator uses the heat pump cycle that consists of a condenser, capillary tube, evaporator, and compressor. In a heat exchanger, cooled air is produced to prevent the spoilage of food in the refrigerator and freezer sections. Figure 1 shows that a refrigerator covers some systems (or modules): the cupboard and door, shelves and boxes, compressor or motor, evaporator and condenser, water supply and ice-making apparatus, controller, and diverse parts. A domestic refrigerator includes as many as 2000 elements. It can be divided into up to 20 units (or 8~10 modules) holding 100 elements each.

If the objective of system life is presumed to have a B20 life 10 years, the lifetime objective of all units should have a B1 life 10 years. If we assume a new subsystem, named Module #3, has a design flaw with a shorter life, it will determine the life of the total refrigerator.

The BX lifetime, $L_B$, could be explained as a quantity of lifetime that X percent of the population has been unsuccessful. This is expressed as "BX life Y years". If the lifetime of a part is B20 life 10 years, 20% of the concerned parts should not fail for ten years. In contrast, the reverse of the failure rate, the B60 life, denotes the mean time to failure (MTTF). The B60 life is not utilized for determining the product life because it is too long for 60% of the systems to fail. The BX lifetime is an acceptable indicator of system life.

### 2.2. Posing an Entire ALT Procedure

Reliability can be described exactly as the potential for a product to perform under a set of prescribed operational/environmental situations for a required period of time [31]. It is often demonstrated as the bathtub (Figure 2). This figure shows three curves. Each curve may be expressed in accordance with the shape parameter in the Weibull chart. In the 1st division, there is a declining rate of failure in the premature section of the system's life ($\beta < 1$). In the 2nd division, there is a uniform rate of failure ($\beta = 1$) in the medium lifetime of the system, pursuing an exponential distribution. Finally, there is a growing rate of failure to the ending of the product life ($\beta > 1$), which follows a Weibull distribution.

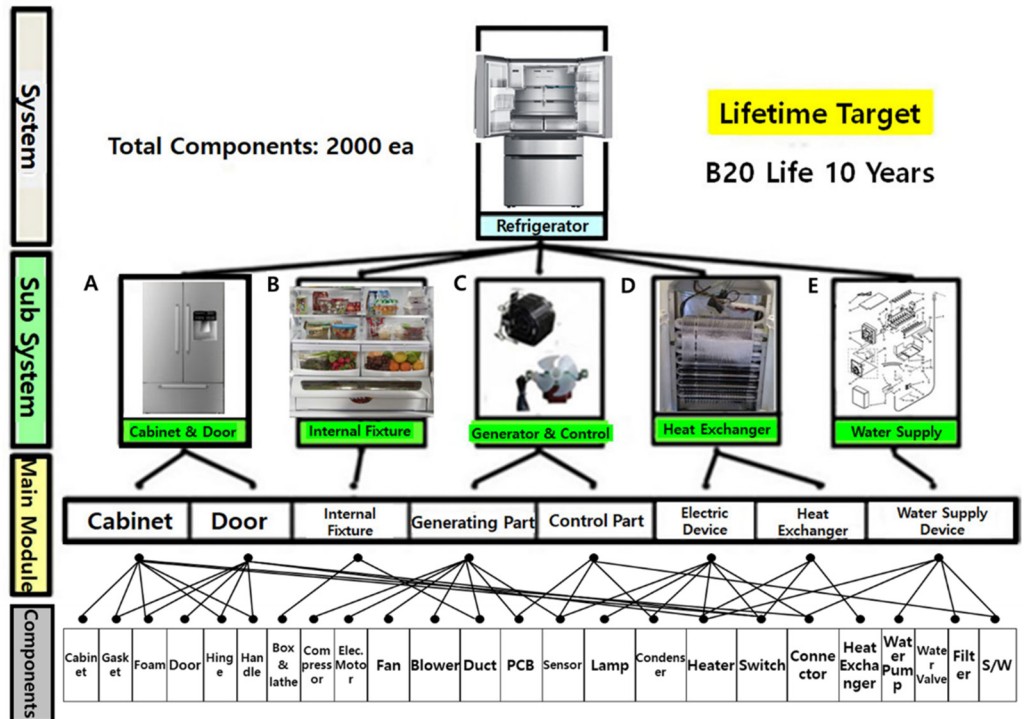

**Figure 1.** Classification of a domestic refrigerator with multiple subsystems.

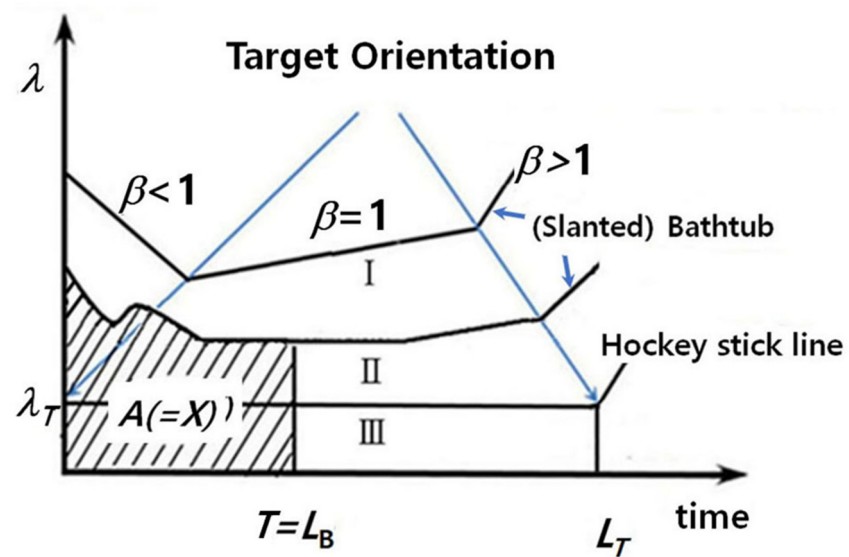

**Figure 2.** BX lifetime ($L_B$) in the (slanted) bathtub.

The unreliability or accumulative distribution function (CDF), $F(t)\,(or\ X) = 1 - R(t)$, is expressed by

$$F(t) = P(T < t) \tag{1}$$

On the (slanted) bathtub in Figure 3, the failure rate, $\lambda$, shall be defined as:

$$\lambda(t) = f(t)/R(t) = \frac{dF(t)/dt}{R(t)} = \frac{d(1 - R(t))/dt}{R(t)} = \frac{-R'(t)}{R(t)} \tag{2}$$

where $f$ is the density function.

If Equation (2) takes the integral, the life of $X\%$ accumulative failure, $F(L_B)$, at $t = L_B$ can be assessed. $F(L_B)$ is expressed as follows:

$$F(t) = \int \lambda(t) dt = \int \frac{-R'(t)}{R(t)} dt = -lnR(t) \tag{3}$$

Or $A(or\ X) = \langle \lambda \rangle \cdot L_B = \int_0^{L_B} \lambda(t) \cdot dt = -lnR(L_B) = -ln(1-F) \cong F(L_B) \tag{4}$

As $T_1$ is presumed to have the period of the 1st failure in the 2nd division of the (slanted) bathtub, reliability, $R(t)$, is expressed as follows:

$$R(t) = P(no\ failure\ in\ (0,t]) = \frac{(m)^0 e^{-m}}{0!} = e^{-m} = e^{-\lambda t} \tag{5}$$

where $m$ is the parameter $(m = \lambda t)$.

As the failure rate of a system mimics the features on the (slanted) bathtub (Stage I or II), it will be unsuccessful in the market. Due to design flaws, a great number of premature failures in the untimely part of the curve could spoil the brand name of the company with the product release. High failure rates in the initial product's lifetime require warranty costs on the manufacturer, and the market share would be anticipated to be negatively affected. The company would be required to enhance the system by (1) removing unpredicted premature failures, (2) lessening (random) failures for its function time, and (3) enlarging the product life.

As a structural design is improved, the system lifetime of the product in the market should increase, and its failure rate should decrease. In this situation, the (slanted) bathtub might be altered to a curve with lower initial failure rates and a longer lifetime. Eventually, as the system is redesigned, the accumulative failure rate, $F(t)$, should improve until the expected design life is met. The product's bathtub shall be similar to the straight line (stage I → III) in Figure 2.

In Equation (5), the product reliability is simply expressed as the failure rate, $\lambda$, and lifetime, $L_B$. Namely,

$$R(L_B) = e^{-\lambda L_B} \cong 1 - \lambda L_B \tag{6}$$

This correlation in Equation (6) is adequate and less than just 20% of the cumulative failure rate [32].

For example, an ice-maker repetitively requires a straightforward mechanical operation: (1) water is provided to the flat and shallow container; (2) it then solidifies into ice by cooled air being blown over it; and (3) the ice is then harvested until the ice container is filled. The ice is retrieved by the consumer when they apply force on a lever that allows the cubed (or crushed) ice to be dispensed. During the process, an ice-maker is subjected to repeated stresses. Failed parts from the field are decisive for comprehending and pinpointing the repetitive usage methods of end-users and identifying structural imperfections in the structure. From the marketplace statistics, the real cause(s) of the troublesome auger motor, including the bearing, was identified. When setting the objective life, $L_B$, by employing an ALT, the part functioned by machine shall be altered by pinpointing the controversial component and improving it (Figure 3).

## Noise Factors

**N1: Customer Usage & Load Conditions**
**N2: Environment Conditions**

Push the lever → Input → Ice-Maker → Response → Harvest Ice

## Control Factors

**C1: AC auger motor including bearing**
**C2: Ice bucket assembly**
**C3: Ice route assembly**
**C4: Lever and controller**

**Figure 3.** Parameter description of the ice-maker (example).

From the market statistics—present lifetime and failure rate—the real cause(s) of the troublesome ice-maker failed from the end-user had been identified. To fulfill the desirable reliability from the objective lifetime, $L_B$, and failure rate, $\lambda$, the possible design flaws of the component might be found and altered by utilizing an ALT.

To reach the target product life by the ALT, three subsystems (or modules) were classified: (1) a modified system, (2) a newly designed system, and (3) the same system. The ice-maker in a household refrigerator utilized in this test investigation was a system which had design defects that should be corrected. End-users had been demanding a replacement ice-maker when the original units had been failing prematurely. Through system D (Table 1) from the market, data had a failure rate of 0.20% per year and a B1 life of 5.0 years. To reply to end-user appeal, an objective life for the ice-maker was specified to have a B1 life 10 years.

**Table 1.** Complete ALT idea of systems in a household refrigerator.

| Modules | Market Reliability | | Predicted Reliability | | | | Goal Reliability | |
|---|---|---|---|---|---|---|---|---|
| | Failure Rate per Year, %/Year | BX Life, Year | Failure Rate per Year, %/Year | | | BX Life, $L_B$ (Year) | Failure Rate per Year, %/Year | BX Life, Year |
| A | 0.34 | 5.3 | New | ×5 | 1.70 | 1.1 | 0.15 | 12(BX = 1.8) |
| B | 0.35 | 5.1 | Same | ×1 | 0.35 | 5.1 | 0.15 | 12(BX = 1.8) |
| C | 0.25 | 4.8 | Modified | ×2 | 0.50 | 2.4 | 0.10 | 12(BX = 1.2) |
| D | 0.20 | 6.0 | Modified | ×2 | 0.40 | 3.0 | 0.10 | 12(BX = 1.2) |
| E | 0.15 | 8.0 | Same | ×1 | 0.15 | 8.0 | 0.10 | 12(BX = 1.2) |
| Miscellaneous | 0.50 | 12.0 | Same | ×1 | 0.50 | 12.0 | 0.50 | 12(BX = 6.0) |
| System | 1.79 | 7.4 | - | - | 3.60 | 3.7 | 1.10 | 12(BX = 13.2) |

### 2.3. Deduction of Life-Stress Model

Because customers wanted to have (cubed or crushed) ice, an ice-maker was developed to be included in a household refrigerator. The major components in a household ice-maker are the geared auger motor, bucket case, helix upper dispenser, blade, etc. An auger motor has two or more gears working together by interlocking their teeth and revolving each other to produce torque and speed. As the motors are engaged, the geared trains lessen the speed of the augers and increase the torque. The auger motor operated by the alternating current (AC) grows the torque by gearbox to break it up at the end of an ice-maker. As a consumer pushes the lever with a cup on the dispenser, (cubed/crushed) ice flows to fill the cup. Consequently, the ice-maker shall be subjected to repeated stresses due to

loading/unloading in the process of crushing ice. If there are structural flaws, such as an inadequate strength to withstand repeated loads, the ice-maker can be successful before satisfying its targeted life. That is, failure happens when the materials in the system parts are too fragile to withstand the exerted stress under environmental circumstances [33].

Reproducing the field failures by ALT, an engineer must understand and quantify the loading that is encountered by the ice-maker in the field before designing the system shape and materials to achieve the objective reliability of the system. Once optimally redesigned, the product might be anticipated to endure the minimum repeated loading in its expected life so that it may extend the targeted life. From the relation between load and lifetime, the (generalized) life-stress prototype that will integrate with geometry and material as the design solution should be derived, which can be described by the phenomena of void generation/transport from the level of quantum mechanics. Eventually, cracks and their propagation might be described by a sample size formulation (Figure 4).

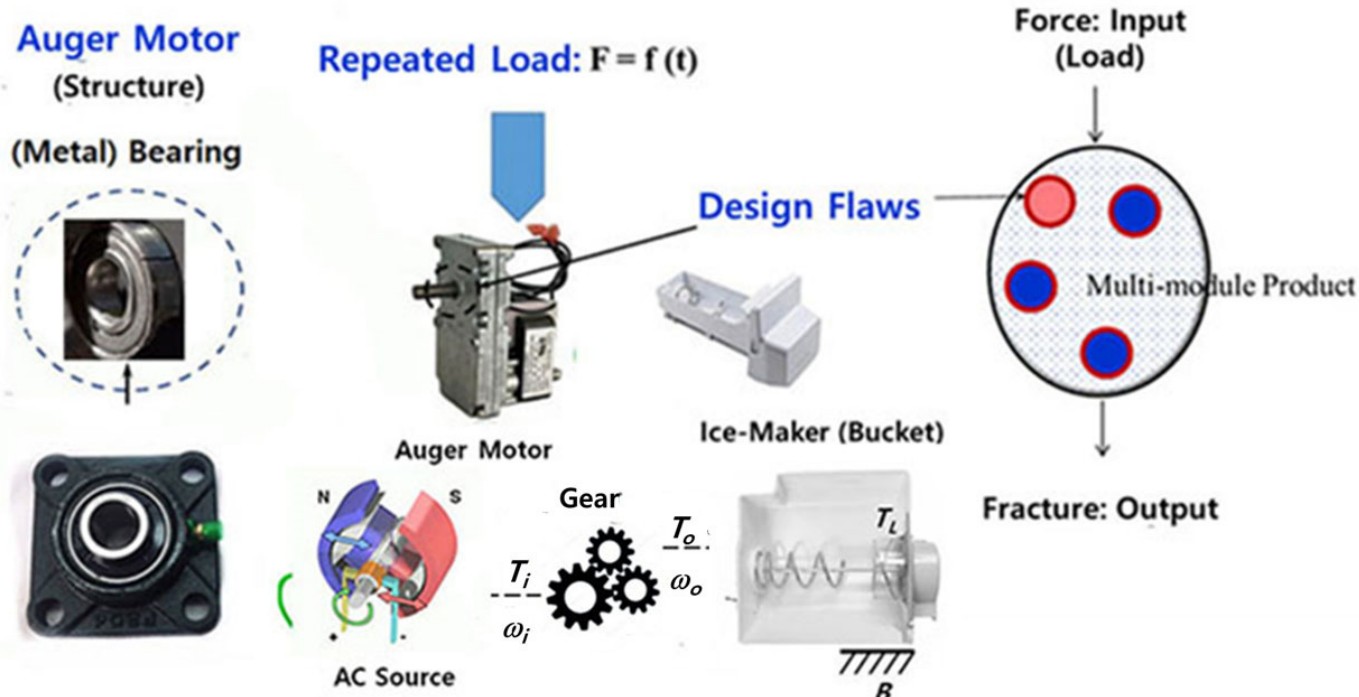

**Figure 4.** Fatigue produced by repeated loads and design deficiencies.

The motivation for the ALT is to resolve how premature the anticipated failure mode might be pinpointed by mathematically employing the work for parametric modeling. That is why elevated tests need to be carried out. To depict the elevated testing time into actual usage time, it is necessary to arrange a straightforward failure expression and resolve the correct numerical method for the life type. The life-stress (LS) type, requiring quantifiable stresses and reaction factors, should be developed. Thus, it will express mechanical failure, such as structural fatigue. Fatigue on the surface of a structure can occur not only due to component stresses but also due to defects such as cracks.

$$\frac{d^2\psi}{dx^2} + K^2\psi = 0 \tag{7}$$

where $K^2 = \frac{8\pi^2 mE}{h^2}$, $E$ is the (electron) energy, $h$ is the Planck constant, and $m$ is the electron mass.

The boundary conditions are as follows: (1) $\psi_n$ limited in the metal but decaying exponentially. The solution of Equation (7) shall be attained [34,35]:

$$\psi(x) = \sqrt{\frac{2}{a}} sin\left(\frac{n\pi}{a}\right) x \text{ or } E_n = \frac{n^2 h^2}{8ma^2} \ n > 0 \tag{8}$$

where $\psi(x+a) = \psi(x)$, $a$ is the (periodic) interval, and $n$ is the main quantum number.

The diffusion phenomena can be expressed (Table 2) [36,37]:

$$J = LD \tag{9}$$

where $J$ is the diffusion flux, $D$ is the driving force, and $L$ is the transport constant.

**Table 2.** Linear transport phenomena.

| | Ohm's Law: $j = -\sigma \nabla V$ | |
|---|---|---|
| $J$ = current density, $j$ (quantity: A/cm$^2$) | $D$ = electric field, $-\nabla V$ (quantity: V/cm, V = potential) | $L$ = conductivity, $\sigma = 1/\rho$ (quantity: $\rho$ = resistivity ($\Omega$ cm)) |
| | Fourier's Law: $q = -\kappa \nabla T$ | |
| $J$ = heat flux, $q$ (quantity: W/cm$^2$) | $D$ = thermal force, $-\nabla T$ (quantity: °K/cm, T = temperature) | $L$ = thermal conductivity, $\kappa$ (quantity: W/°K cm) |
| | Fick's Law: $F = -D\nabla C$ | |
| $J$ = material flux, $F$ (quantity:/sec cm$^2$) | $D$ = diffusion force, $-\nabla C$ (quantity:/cm$^4$, C = concentration) | $L$ = diffusivity, $D$ (quantity: cm$^2$/sec) |
| | Newton's Law: $F_u = -\mu \nabla u$ | |
| $J$ = fluid velocity flux, $F_u$ (quantity:/sec$^2$ cm) | $D$ = viscous force, $-\nabla u$ (quantity:/sec, u = fluid velocity) | $L$ = viscosity, $\mu$ (quantity:/sec cm) |

In particular, when an electromagnetic force, $\xi$, is exerted, the metal impurities, caused by electronic movement, easily float to the right-hand as the junction magnitude energy is lowered. Expressing the solid-state diffusion of impurities of silicon in a semiconductor can be shortened: (1) electromigration-induced voiding; (2) build-up of chloride ions; and (3) trapping of electrons or holes. The transport diffusion process, $J$, might be defined as [38]:

$$J_1 = [aC(x-a)] \cdot exp\left[-\frac{q}{kT}\left(W - \frac{1}{2}a\xi\right)\right] \cdot v \tag{10}$$

where $[aC(x-a)]$ is the density per unit area of electric particles located in the valley at $(x-a)$, and the exponential factor is the chance of a successful jump from the valley at $(x-a)$ to the valley at $x$. Note the lowering of the barrier due to the electric field $\xi$.

Similar formulas can be expressed for $J_1$, $J_2$, and $J_3$. As these are integrated to formulate the flux $J$ at position $x$, with the concentration $C(x \pm a)$ approximately by $C(x) \pm a(\partial C/\partial x)$, the flux $J$ is

$$J = -\left[a^2 v e^{-qw/kT}\right] \cdot cosh \frac{qa\xi}{2kT} \frac{\partial C}{\partial x} + \left[2ave^{-qw/kT}\right] C \, sinh \frac{qa\xi}{2kT} \tag{11}$$

where $C$ is the concentration quantity, $q$ is the amount of accumulated electrical energy, $v$ is the jump frequency, $a$ is the atomic intervening time, $\xi$ is the exerted field, $k$ is Boltzmann's quantity, and $T$ is the (absolute) temperature.

Unless the electric field is relatively small, i.e., $\xi \ll \frac{qa}{2kT}$, Equation (11) might be redefined as follows:

$$J = \Phi(x,t,T) sinh(a\xi) exp\left(-\frac{Q}{kT}\right) \tag{12}$$

or

$$J = B \, sinh(a\xi)exp\left(-\frac{Q}{kT}\right)$$ (13)

where $Q$ is the energy and, $\Phi()$ and $B$ are constants.

If Equation (13) captures a reverted formulation, the life-stress (LS) type shall be clarified as:

$$TF = A[sinh(aS)]^{-1}exp\left(\frac{E_a}{kT}\right)$$ (14)

For a prototype, Equation (14) is clarified as a general expression because the sine hyperbolic expression $[sinh(aS)]^{-1}$ designating stress shall be exchanged into a power (or exponential) formulation. It then may outline most of the LS prototypes about some failure, such as fatigue in the system. It can be conveyed as follows: (1) first, $(S)^{-1}$ has a nearly straight line; (2) second, $(S)^{-n}$ has what is viewed; and (3) third, $(e^{aS})^{-1}$ is largely developed (Figure 5).

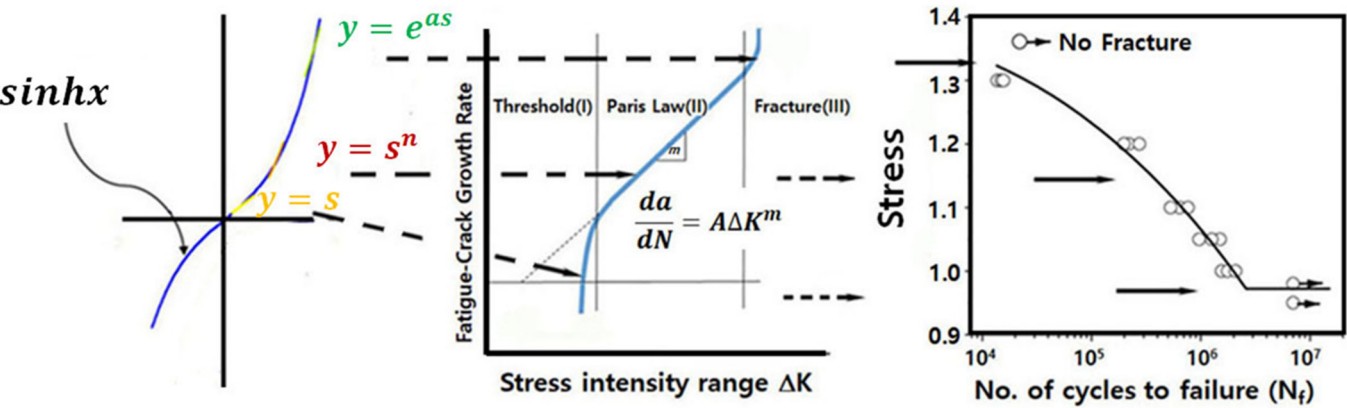

**Figure 5.** Denotation of the hyperbolic sine stress in the Paris law and S–N curve.

Because ALT is frequently carried out in the span amidst stress, Equation (14) is expressed as:

$$TF = A(S)^{-n}exp\left(\frac{E_a}{kT}\right)$$ (15)

where $n = -\left[\frac{\partial ln(TF)}{\partial ln(S)}\right]_T$.

For an expressed crack and structural form, Equation (15) can be redefined as

$$TF = B(\Delta K)^{-n}exp\left(\frac{Q}{kT}\right)$$ (16)

where $B$ is constant, $\Delta K = YS(or \, \Delta\sigma)\sqrt{\pi a}$.

As the stress intensity component, $\Delta K$, is exerted on a material, the crack will produce to a specific amount $\Delta a$, which relies on the crack growth speed, $\Delta a/\Delta N$, in component shapes such as crack tips such as grooves, slender areas, holes, etc. It therefore propagates to a risky magnitude. As loads are exerted until the targeted lifetime, $L_B$ or mission time, the stress raisers (or material) in a component can be discovered.

The stress of a product functioned by machinery is a complex quantity to formulate in a raised testing. Because the energy is clarified as the product of flow and effort, the stress comes from effort in an energy transport system [39]. Thus, Equation (15) or (16) can be stated as follows:

$$TF = A(S)^{-n}exp\left(\frac{E_a}{kT}\right) = C(e)^{-\lambda}exp\left(\frac{E_a}{kT}\right)$$ (17)

where $C$ is constant.

The acceleration factor (*AF*) is clarified as the portion between the raised stress and typical functioning situation. From Equation (17), *AF* shall be modified to merge the effort idea:

$$AF = \left(\frac{S_1}{S_0}\right)^n \left[\frac{E_a}{k}\left(\frac{1}{T_0} - \frac{1}{T_1}\right)\right] = \left(\frac{e_1}{e_0}\right)^\lambda \left[\frac{E_a}{k}\left(\frac{1}{T_0} - \frac{1}{T_1}\right)\right] \tag{18}$$

*2.4. Obtaining of the Sample Size Formulation for ALT*

To accomplish the desired assignment time of the ALT from the targeted BX life in the testing plan, expressed in Sections 2.1 and 2.2, the sample size formulation integrated with AF in Section 2.3 might be derived. The Weibull distribution for system life is extensively employed because it is defined as an expression of the characteristic life, $\eta$, and shape parameter, $\beta$. Therefore, if the system keeps to the Weibull distribution, the accumulative failure rate, $F(t)$, in Equation (1) is defined:

$$F(t) = 1 - e^{-\left(\frac{t}{\eta}\right)^\beta} \tag{19}$$

where *t* is the (passed) time.

If Equation (19) takes the logarithm at $t = L_B$, it is

$$L_B^\beta = ln(1-x)^{-1}\eta_\alpha^\beta \tag{20}$$

where *x* is the accumulative failure rate until lifetime ($x = F/100$), and $\eta_\alpha^\beta$ is the characteristic life.

Failures on the Weibull distribution are divided into some classes—infant mortality, random failure, and wear-out failure—depending on shape parameter (see Figure 2). The Weibayes procedure is explained as a Weibull examination with an assigned shape parameter, which can be obtained from prior experience or test data. Mentioning on Weibayes, characteristic life, $\eta_{MLE}$, is attained from utilizing the maximum likelihood estimate (MLE):

$$\eta_{MLE}^\beta = \sum_{i=1}^n t_i^\beta / r \tag{21}$$

where $t_i$ for each sample is testing time, and *r* is the failure numbers.

The confidence level is $100(1-\alpha)$, so characteristic life, $\eta_\alpha$, can be assessed as follows:

$$\eta_\alpha^\beta = \frac{2r}{\chi_\alpha^2(2r+2)} \cdot \eta_{MLE}^\beta = \frac{2}{\chi_\alpha^2(2r+2)} \cdot \sum_{i=1}^n t_i^\beta \tag{22}$$

If Equation (22) is substituted into Equation (20), it is

$$L_B^\beta = ln(1-x)^{-1}\frac{2}{\chi_\alpha^2(2r+2)} \cdot \sum_{i=1}^n t_i^\beta \tag{23}$$

As the whole reliability test is carried out with a limited sample number, the test scheme can be defined as:

$$nh^\beta \geq \sum t_i^\beta \geq (n-r)h^\beta \tag{24}$$

If Equation (24) is inserted into Equation (23), it is

$$L_B^\beta \geq ln(1-x)^{-1}\frac{2}{\chi_\alpha^2(2r+2)} \cdot (n-r)h^\beta \geq L_B^{*\beta} \tag{25}$$

If Equation (25) is reordered, the sample size expression is found as:

$$n \geq \frac{\chi_\alpha^2(2r+2)}{2} \times \frac{1}{ln(1-x)^{-1}} \times \left(\frac{L_B^*}{h}\right)^\beta + r \tag{26}$$

As the 1st term $\frac{\chi_{\alpha}^2(2r+2)}{2}$ in a 60% confidence level is approximated to $(r+1)$ and $ln\frac{1}{1-x}$ approximates to $x$, Equation (26) is redefined as:

$$n \geq (r+1) \times \frac{1}{x} \times \left(\frac{L_B^*}{h}\right)^{\beta} \tag{27}$$

As *AF* in Equation (18) is replaced into the test time, *h*, Equation (27) shall be redefined:

$$n \geq (r+1) \times \frac{1}{x} \times \left(\frac{L_B^*}{AF \cdot h_a}\right)^{\beta} \tag{28}$$

where Equation (28) will be clarified as $n \sim$ (failed samples + 1)·(1/cumulative failure rate)·((objective life/(test time)) ^ $\beta$.

Equation (28) shall be affirmed as [1,40] and Appendix A. Namely, for $n \gg r$, the sample size shall be expressed as:

$$n = \frac{\chi_{\alpha}^2(2r+2)}{2m^{\beta}lnR_L^{-1}} = \frac{\chi_{\alpha}^2(2r+2)}{2} \times \frac{1}{ln(1-F_L)^{-1}} \times \left(\frac{L_B}{h}\right)^{\beta} \tag{29}$$

where m $\cong h/L_B$.

If $r = 0$, the sample size can be indicated as:

$$n = \frac{ln(1-CL)}{m^{\beta}lnR_L} = \frac{-ln(1-CL)}{-m^{\beta}lnR_L} = \frac{ln(1-CL)^{-1}}{m^{\beta}lnR_L^{-1}} = \frac{ln\alpha^{-1}}{m^{\beta}lnR_L^{-1}} = \frac{\chi_{\alpha}^2(2)}{2} \times \frac{1}{ln(1-F_L)^{-1}} \times \left(\frac{L_B}{h}\right)^{\beta} \tag{30}$$

where $2ln\alpha^{-1} = \chi_{\alpha}^2(2)$ and *CL* is the confidence level.

If the objective of a product life–ice-maker is presumed to have a B1 life 10 years, the allocated test is computed for the assigned parts under raised circumstances. In executing parametric ALTs, the structural defects of a product operated by machinery will be found and altered to obtain the intended system life.

*2.5. Case Investigation—Magnifying Life of an Ice-Maker Incorporating Auger Motor with a Bearing in a Household Refrigerator*

Because customers want the convenience of (cubed or crushed) ice being distributed from a household refrigerator, an ice-making system was designed in a refrigerator. As a consumer utilizes a cup to apply force on the lever, ice is dispensed. The major parts include an auger motor with a geared system and bearings, helix upper dispenser, etc. These components must be designed for high-strength fatigue because of the repetitive stresses they are subjected to under normal consumer usage (Figure 6).

In ice-making, the parts undergo repeated mechanical loads and need to be strong enough to not fracture due to fatigue before the expected life. A household refrigerator in the United States is designed to harvest ice at a rate of 10 cubes per use and 200 cubes per day. Ice harvesting may also be affected by individual end-user usage patterns, such as ice usage, (tap) water pressure, notch positions in refrigerators, and the cycles of door opening. When set to the crushed mode, the ice-maker is repetitively subjected to (impact) loads in crushing ice. In the market, ice-makers, including auger motors, were unsuccessful under unidentified consumer usage in a refrigerator. Field statistics also showed that the ice-makers returned from the market had structural defects such as material problems (high carbon alloy steel with 1.30–1.65% chromium) under the typical freezing temperatures (below −20 °C) found in the refrigerator. For the customer, the ice-maker system experienced a sudden failure and no longer functioned. Engineers were required to determine the basic causes by a failure analysis (or laboratory tests) and then modify the ice-maker (Figure 7).

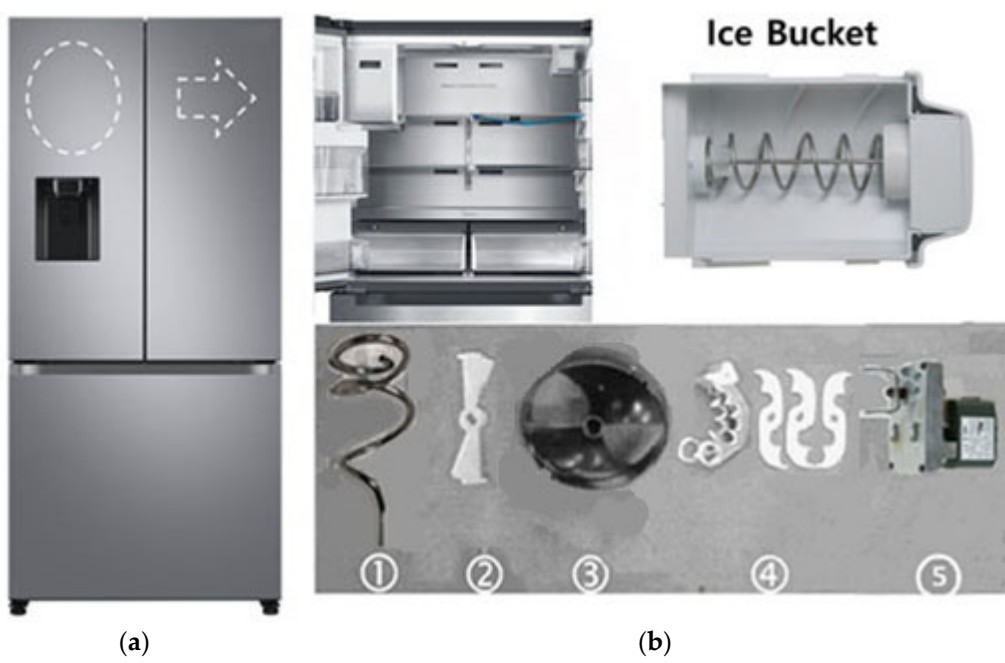

**Figure 6.** A household refrigerator equipped with ice-maker. (**a**) Household refrigerator; (**b**) components of a domestic ice-maker: helix support ①, blade dispenser ②, helix upper dispenser ③, blade ④, and auger motor ⑤.

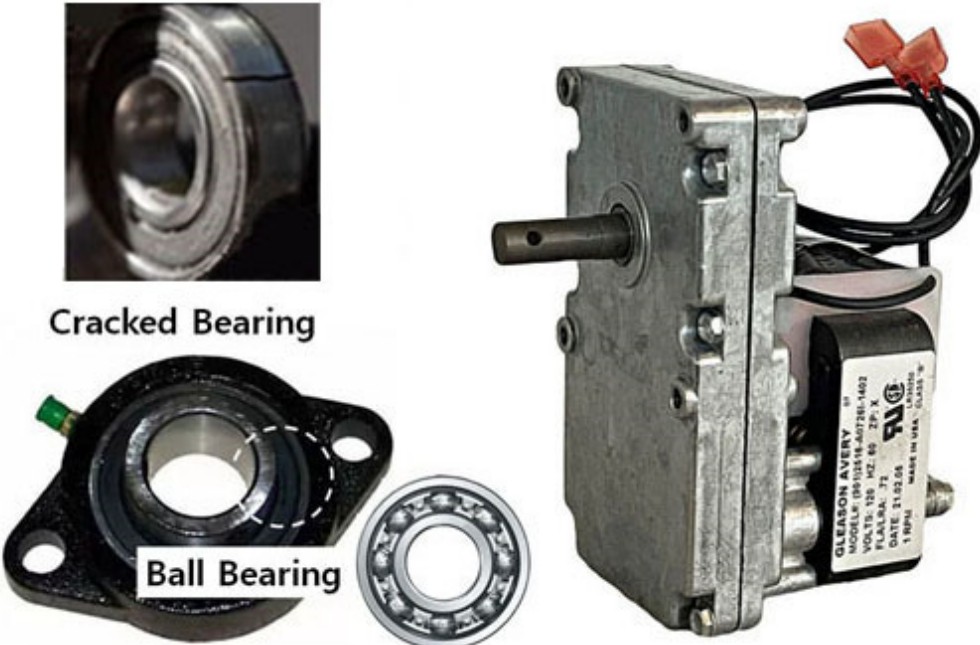

**Figure 7.** Failed auger motor in the market.

By utilizing a failure analysis (and laboratory tests) for failed market parts, under typical freezer temperatures (below −20 °C), a crack began in the outer ring of the bearing and propagated to the end. The system had to be redesigned. If there are structural defects—improper bearing material in the auger motor—where repeated loads are exerted in the freezer section, it will fail before its expected life. To reproduce the troublesome component(s) and modify them, a designer was required to perform the ALT for a product. It consisted of (1) a load inspection for the troublesome product (Section 2.5), (2) the measures

of taking the feasible and actual usage of ALTs with modifications (Section 3), and (3) the appraisal of whether the life target of current structures (Section 3) had been accomplished.

To attain the differential equations which are made up of state variables for the parametric prototype, the bond graph in Figure 8 shall be settled as follows (Appendix B):

$$\begin{bmatrix} di_a/dt \\ d\omega/dt \end{bmatrix} = \begin{bmatrix} -R_a/L_a & 0 \\ mk_a & -B/J \end{bmatrix} \begin{bmatrix} i_a \\ \omega \end{bmatrix} + \begin{bmatrix} 1/L_a \\ 0 \end{bmatrix} e_a + \begin{bmatrix} 1 \\ -1/J \end{bmatrix} T_L \tag{31}$$

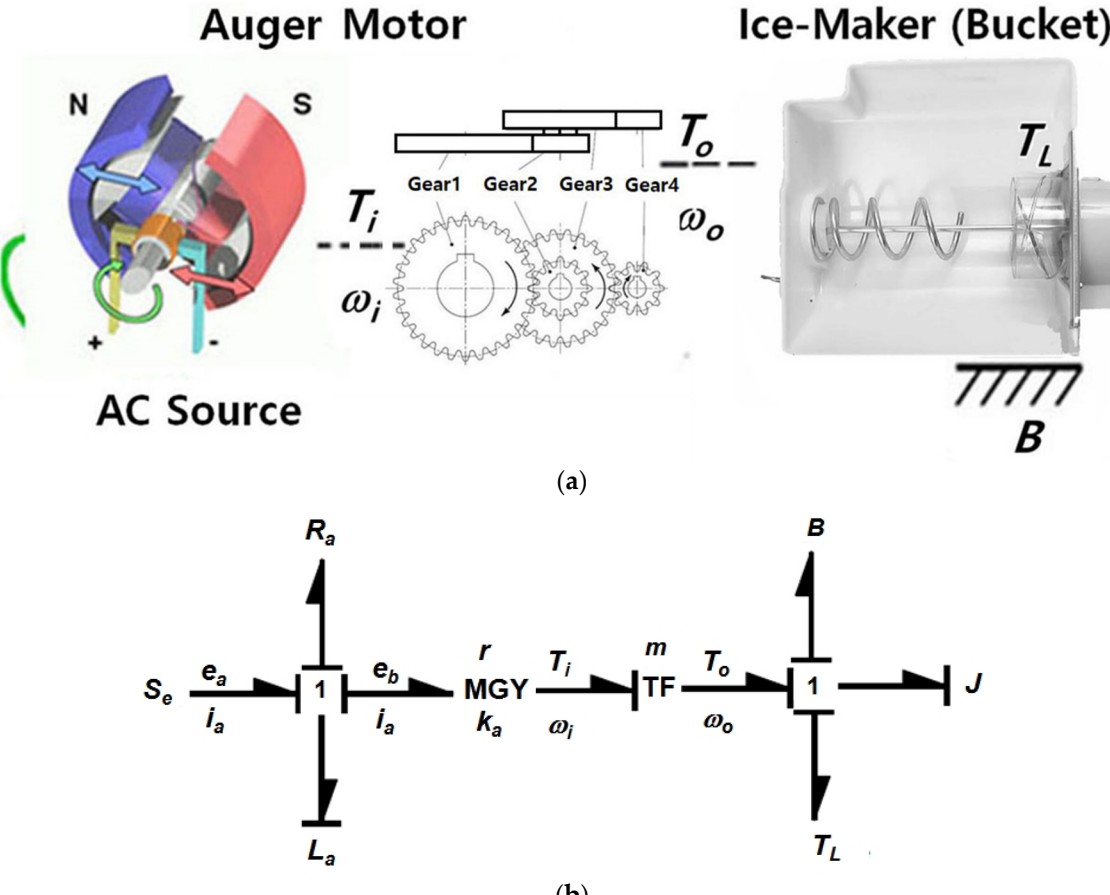

**Figure 8.** Design concepts of an ice-maker. (**a**) Exemplifying drawing of the auger motor, ice crusher, bucket, blade, etc. (**b**) Bond graph of an ice-maker.

As the differential equation, Equation (31) finds the integral, the output, $y_p$, harvested by the ice-maker obtained as follows:

$$y_p = \begin{bmatrix} 0 & 1 \end{bmatrix} \begin{bmatrix} i_a \\ \omega \end{bmatrix} \tag{32}$$

From Equation (31), the lifetime of the ice-maker depends on the required torque to harvest the crushed ice. By altering the torque, the ALT can be performed. The life-stress prototype in Equation (17) is adjusted as

$$TF = A(S)^{-n} = AT_L^{-\lambda} = A(F_c \times R)^{-\lambda} = B(F_c)^{-\lambda} \tag{33}$$

where *A* and *B* are constants.

Thus, the *AF* in Equation (18) shall be defined as

$$AF = \left(\frac{S_1}{S_0}\right)^n = \left(\frac{T_1}{T_0}\right)^\lambda = \left(\frac{F_1 \times R}{F_0 \times R}\right)^\lambda = \left(\frac{F_1}{F_0}\right)^\lambda \tag{34}$$

The ALT from Equation (34) can be carried out until the mission time which satisfies the life target—B1 life 10 years—is attained.

The surrounding conditions of an ice-maker in a household refrigerator can change and range from −15 to −30 °C with humidity ranging from 0% to 20%. Depending on the end-user use operating conditions, an ice-maker is expected to operate from three to eighteen cycles per day. For the highest utilization for ten years, the ice-maker can be expected to experience 65,700 life cycles.

The stress amount for the ALT was determined using the permitted utilization range provided from the auger motor manufacturers in bench-marked statistics. Step-stress lifetime testing was applied under the usage conditions for several elevated loads including 0.8 kN-cm, 1.0 kN-cm, and 1.47 kN-cm [41]. These torques were different from the expected applied torque of 0.69 kN-cm in the field. With the higher torques, the failure time of an auger motor at specific stress quantities could be expected to be faster than failures in the field.

Engineering statistics from the auger motor company showed that the common torque was 0.69 kN-cm and the maximum torque was 1.47 kN-cm. If the cumulative damage factor, $\lambda$, was 2, AF in Equation (34) was almost five.

For a life objective, a B1 life 10 years, the number of assignment cycles for ten components (attained by employing Equation (28)) was 42,000 cycles if the shape parameter was assumed to be 2.0. The ALT was set to assure a life objective if it was unsuccessful for 42,000 cycles. Figure 9 shows the test equipment of an ALT for replicating the failed ice-maker, involving the auger motor in the market. Figure 10 shows the duty cycles applied by the crushing torque $T_L$.

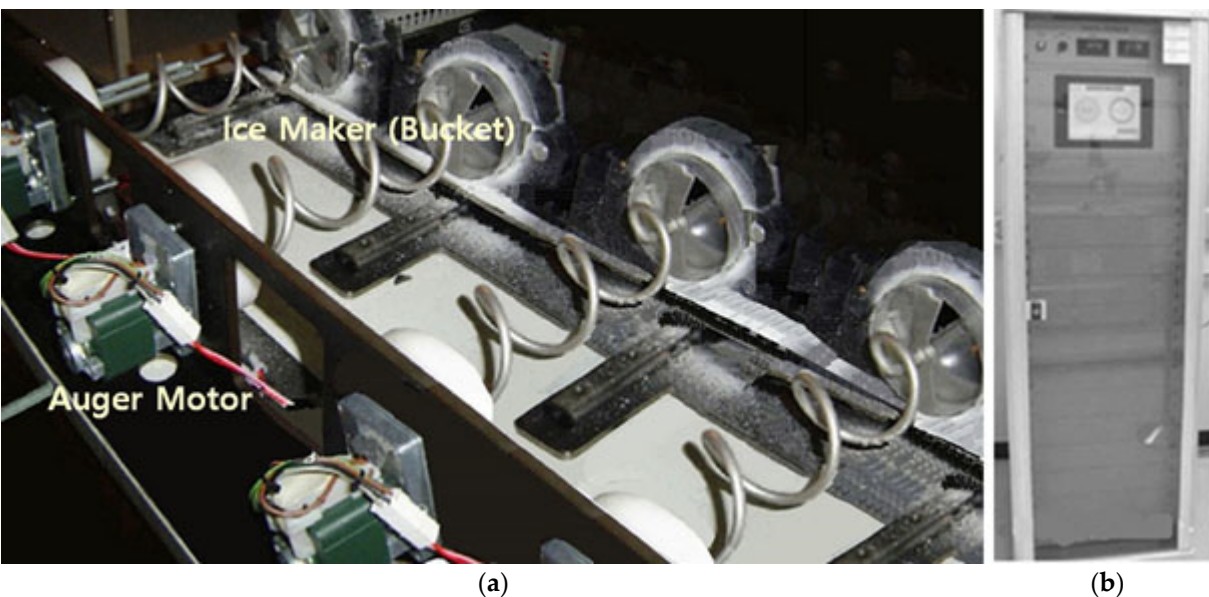

**(a)**            **(b)**

**Figure 9.** ALT system; **(a)** equipment; **(b)** controller.

The evaluated life $L_B$ in every ALT stage is expressed as

$$L_B^\beta \cong x \cdot \frac{n \cdot (h_a \cdot AF)^\beta}{r+1} \tag{35}$$

where $h_a$ is the real testing time.

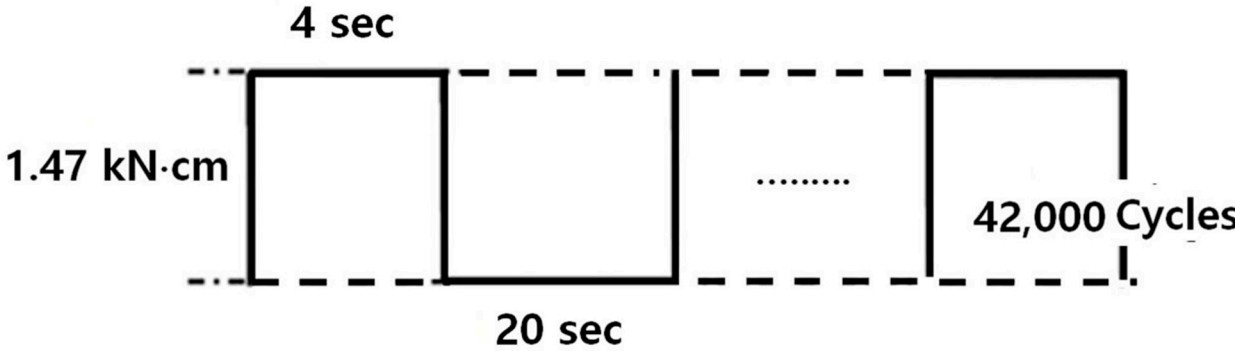

**Figure 10.** Duty cycles applied by ice-squashing torque $T_L$ exerted in the band clamper.

Let $x = \lambda \cdot L_B$. The approximated failure rate $\lambda$ of the selected parts is expressed as

$$\lambda \cong \frac{1}{L_B} \cdot (r+1) \cdot \frac{L_B^{\beta}}{n \cdot (h_a \cdot AF)^{\beta}} \tag{36}$$

In every ALT stage, by measuring the approximated $L_B$ life and failure rate $\lambda$, the reliability of the system design operated by machinery can be secured.

### 3. Results and Discussion

From Equation (33), it can be seen that the life of an ice-maker relies on the exerted torque $T_L$. To rapidly identify the failure time of an ice-maker, the torque was enhanced from Equation (34). Putting a scale of stress level due to the applied load by the step-stress life tests, the failure time(s) were investigated at successive stress quantities: 0.8 kN-cm, 1.0 kN-cm, and 1.47 kN-cm (torque for ALT). For 0.8 kN-cm, the ice-maker terminated at approximately 11,000 cycles. For 1.0 kN-cm, the ice-maker terminated at approximately 9000 cycles and 13,000 cycles. For 1.47 kN-cm, the ice-maker terminated at approximately 5000 cycles and 8000 cycles. Therefore, the stress quantity of 1.47 kN-cm was set for the ALT as it had a comparatively excellent linearity in the Weibull chart, compared to the dissimilar stress quantities.

In the 1st ALT, the fractured bearings of the auger motor occurred at approximately 6500 cycles and 6900 cycles from the failed ice-makers that were disassembled. Figure 11 shows a photograph contrasting with the product returned from the market and that from the 1st ALT. By employing scanning electron microscopy (SEM), the fractures in the images occurred in the outer ring. The fractured surface on the cross-section had an intergranular (IG) crack and fatigue due to repeated impact under severe environmental conditions.

Because failed samples were indistinguishable in shape through the ALT, we might reproduce the fractured outer ring of a bearing in the market. There was a material design defect—AISI 52100 Alloy Steel with 1.30–1.60% chromium—which could not endure the stresses due to the repetitive torques at the freezing temperatures ($-20\,^{\circ}$C below) in the refrigerator. As the bearing and shaft in an ice-maker repetitively struck together, they began to crack and ultimately fractured because the material (AISI 52100 Alloy Steel) was too brittle under these circumstances: repeated impacts and severe cold temperature ($-20\,^{\circ}$C $\downarrow$). Figure 12 shows the visual inspection of the parametric ALT outcomes and field data in the Weibull chart. The shape parameter in the 1st ALT which relied on loading circumstances was estimated to be 2.0. For the test, it was affirmed to be 4.38 in the Weibull chart.

To withstand the repetitive impact torque, the material of the troublesome bearing in an auger motor was modified from AISI 52100 Alloy Steel with 1.30–1.65% chromium to the lubricated sliding bearing with sintered and hardened steel (FLC 4608-110HT). The quantities of AF and $\beta$ in Equation (34) and Figure 12 were confirmed to be 5.0 and 4.1, respectively. Based on the test statistics, because the life target of a new sample was a B1 life

ten years, the test time recomputed in Equation (29) for the ten samples was 23,400 cycles, which would be the statement of the parametric ALT.

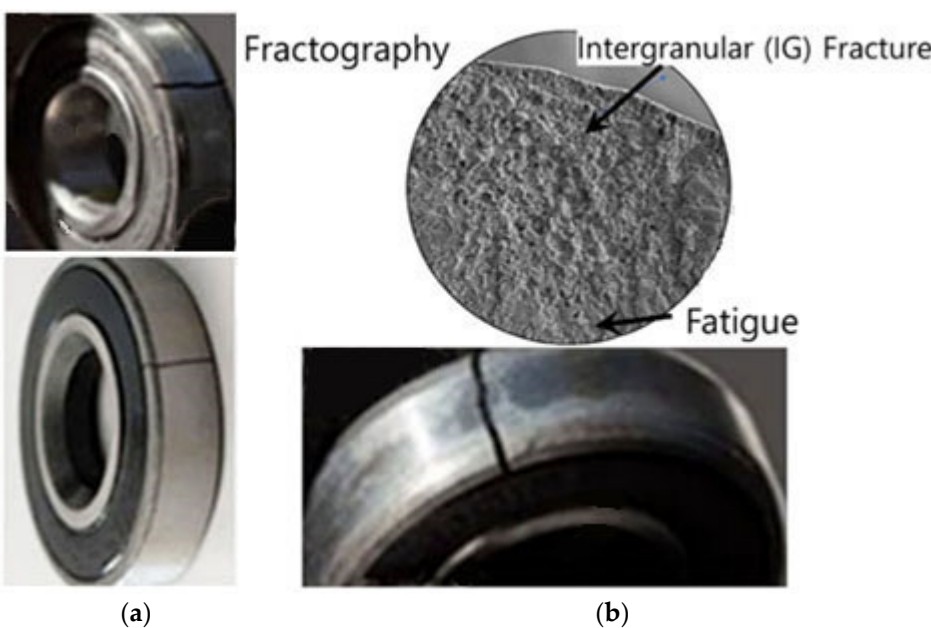

| (a) | (b) |
|---|---|

**Figure 11.** Failed bearing in auger motor from the market and in the 1st ALT; (**a**) unsuccessful part from the marketplace; (**b**) failed bearing within 1st ALT, crack origin at outer ring.

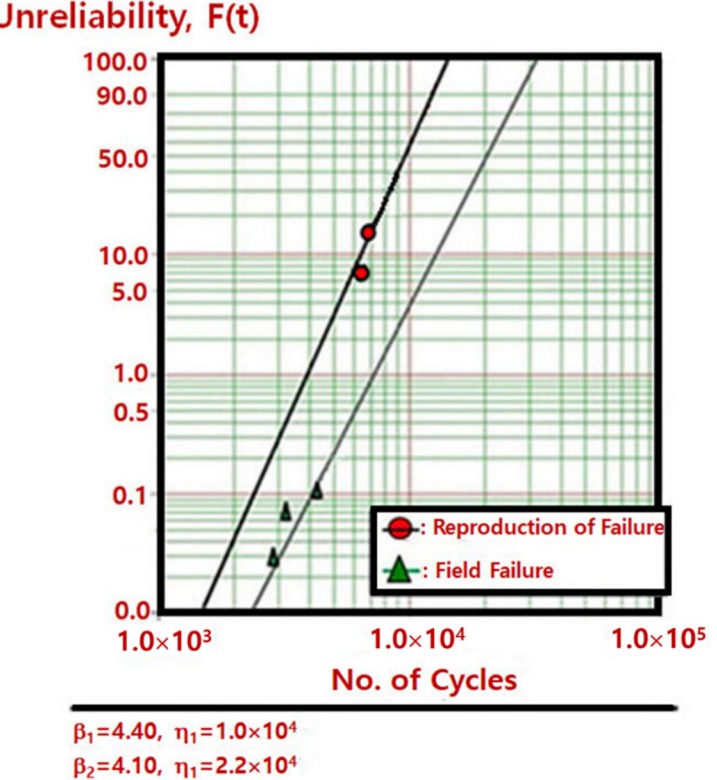

$\beta_1=4.40$, $\eta_1=1.0\times10^4$
$\beta_2=4.10$, $\eta_1=2.2\times10^4$

**Figure 12.** Failed products on the Weibull chart.

In the 2nd ALT, at 10,000 cycles and 12,000 cycles, the fracture of the helix upper dispenser (polycarbonates (PC)) happened in the exposure area of the blade dispenser (Figure 13). To understand the basic cause of the failed product, it was examined. It was discovered that there was a constructional flaw—the weld line between the helix

upper dispenser and the blade dispenser—that had countless microvoids generated in the plastic injection procedure. As the blade dispenser (stainless-steel) stroked the helix upper dispenser (PC) under extreme freezing circumstances, it began to crack and eventually fractured at the weld line (Figure 13b). As an alternative, a strengthened rib of the helix was thickened after the plastic injection procedure was altered. Then, a finite element analysis (FEA), integrated with the ALT, was carried out. As the helix upper dispenser was fastened against the barrier, a simple impact torque (1.47 kN-cm), as shown in Figure 8, was applied. Utilizing materials and processing circumstances close to those of the helix upper dispenser, the constitutive material properties such as PC (helix upper dispenser) were discovered. As a consequence, the stress of the parts by the FEA inspection was lessened from 45.0 kPa to 20.0 kPa (Figure 14) [42,43].

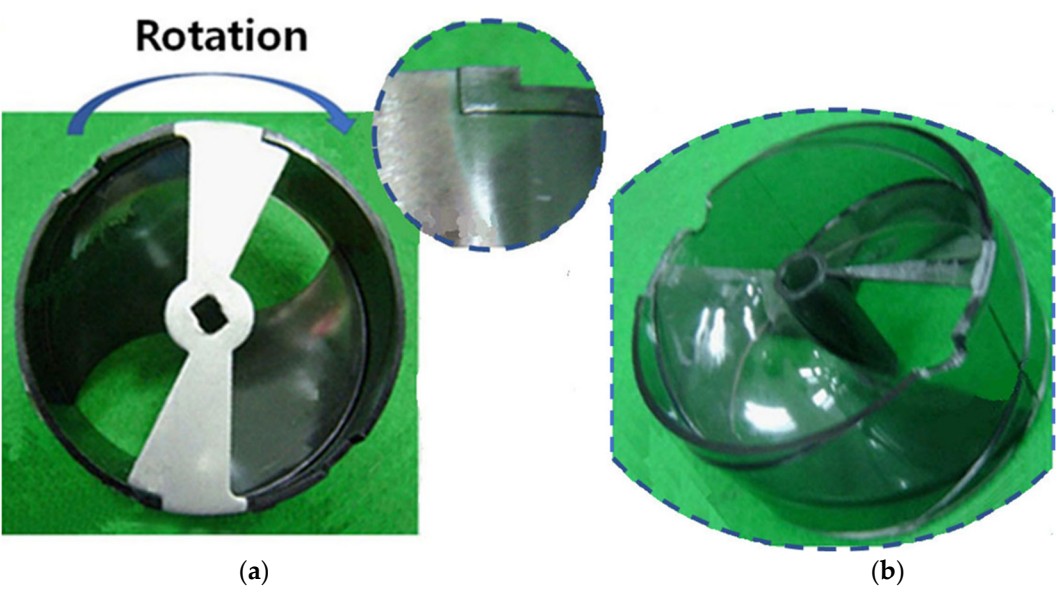

(**a**)  (**b**)

**Figure 13.** Failed parts in the 2nd ALT: (**a**) the basic cause of fractured parts; (**b**) failed parts.

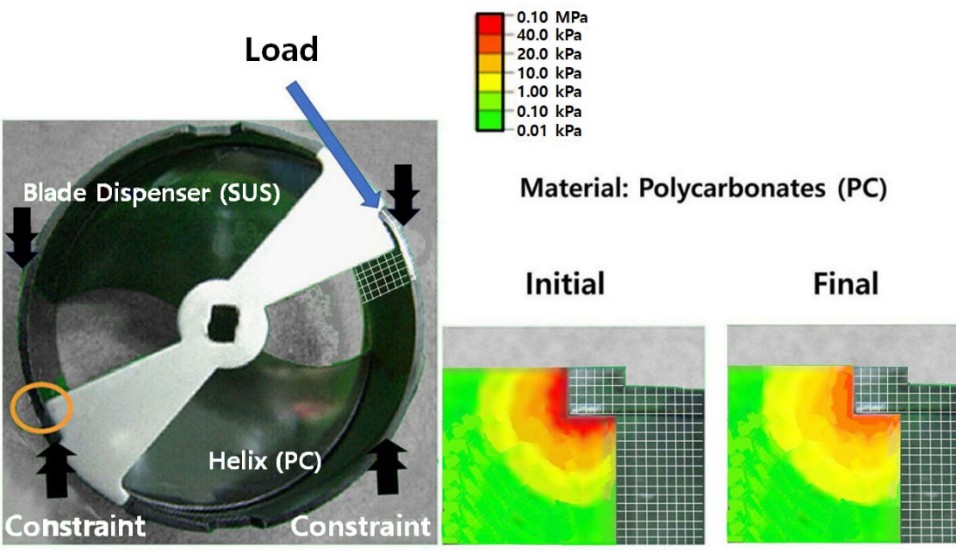

**Figure 14.** Outcome of stress by employing finite element analysis (FEA).

As the material of the bearing in an auger motor was modified and the strengthened rib of the helix upper dispenser was thickened, the lifetime of an ice-maker was expanded. However, as the 23,400 mission time in the 2nd ALT was not yet achieved, the 3rd ALT was performed to assure the structural alternation of an ice-maker.

There was no problem in the 3rd ALT until 60,000 cycles. Over the rounds of three ALTs with alternations, the ice-maker was assured to have B1 life 10 years with an accumulated failure rate of 1% from Equations (35) and (36) when the real cycles, $h_a = 60,000$ cycles; in inserting in lifetime target, $x = 0.01$; sample size, $n = 10$; accelerated factor, $AF = 5.0$; shape parameter, $\beta = 4.10$; and failure number, $r = 0.0$. Table 3 shows a curtailed outcome of the parametric ALTs.

**Table 3.** ALT results for ice-makers.

| Parametric ALT | 1st ALT | 2nd ALT | 3rd ALT |
|---|---|---|---|
| | **Draft Design** | **-** | **Final Design** |
| Over the route of 23,400 cycles, the ice-maker system has no issues | 6500 cycles: 1/10 fail<br>6900 cycles: 1/10 fail<br>(Unsuccessful bearing samples) | 10,000 cycles: 1/10 fail<br>12,000 cycles: 1/10 fail<br>(Unsuccessful helix samples) | 23,400 cycles: 10/10<br>42,000 cycles: 10/10<br>60,000 cycles: 10/10<br>OK |
| Structure | 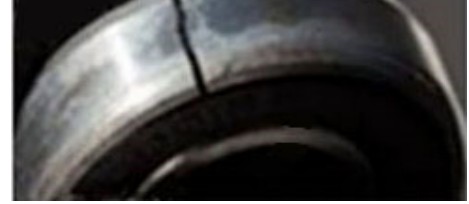 | 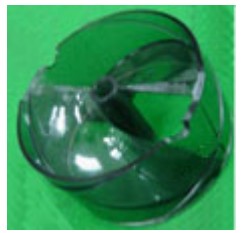 | |
| Action plans | C1: from AISI 52100 Alloy Steel to lubricated sliding bearing with sintered and hardened steel (FLC 4608-110HT) | C2: thickened reinforced rib on the side of helix | |

## 4. Summary

To improve the lifetime of a refrigerator with an ice-maker, a reliability structured method was employed. The method was used to improve the ice-maker augur motor and bearing. The method involved a life-stress prototype by a quantum-transported procedure and a sample size formulation. It covered the following: (1) the product BX life formed the ALT strategy, (2) ALTs with alterations, and (3) resolving if the product design obtained the targeted time. The ice-maker was examined as a case investigation.

- In the 1st ALT, the auger motor in an ice-maker terminated near 6600 cycles and 6900 cycles as exerted for torque $-1.47$ kN-cm under the freezing temperatures ($-20\,^\circ$C below) in the refrigerator. After disassembling the troublesome samples, we found the fractured outer ring of the bearing in an auger motor. As an action plan, the bearing matter in an ice-maker was altered from AISI 52100 Alloy Steel with 1.30–1.60% chromium to lubricated sliding bearing with sintered and hardened steel (FLC 4608-110HT).
- In the 2nd ALT, the helix (polycarbonates) at 10,000 cycles and 12,000 cycles was fractured because the ice-maker had insufficient fatigue strength for repetitive stress in the freezer department. As an alternative, a reinforced rib of the helix was thickened after the plastic injection procedure was modified.
- In the 3rd ALT, no problems were found. The ice-maker, involving an auger motor, will satisfy the life objective of B1 life 10 years. Inspecting controversial field parts and carrying out ALTs with modifications could grow the life of an ice-maker, involving an auger motor with a bearing.
- By identifying and understanding the design problems of products that failed in the market, it was possible to carry out parametric ALTs with system modifications. When reproducing the market failures, the design defects can be recognized and altered. Ultimately, we approximated whether the system reached the life targets. In the process, the life-stress (LS) type and sample size were also utilized.

## 5. Conclusions

As design flaws are identified and corrected, this procedure with reliability quantitative (RQ) specifications will help improve sustainability and fulfill "zero waste" from poorly designed products with regard to the development of new products in the field. It has been relevant to different products operated by machinery such as agricultural machines, construction machines, airplanes, automobiles, etc. Engineers also are required to grasp why multimodule systems fail in their life. Namely, if there are structural defects where it is repeatedly subjected to loading in its working, the product shall be unsuccessful during its expected life. Preventing the fabrication of troublesome material (AISI 52100 Alloy Steel) in a mechanical system will help transform the lifestyle by properly utilizing a newly developed material in the agroindustry for the circular economy and preventing influencing the environmental change worldwide. Furthermore, customers will use a product of good quality and prevent generating futile waste. Moreover, this innovative and computational study will enhance the individual lifestyle in economics and permit forecasting the assessment of global transformation and evolution associated with sustainability.

**Author Contributions:** S.W. carried out the idea development, method, investigation, and experiment, and wrote the manuscript. D.L.O. edited the original documents and provided comments on the methods. Y.M.H. and G.M. altered the draft. All authors have read and agreed to the published version of the manuscript.

**Funding:** This research received no external funding.

**Institutional Review Board Statement:** Not applicable.

**Informed Consent Statement:** Not applicable.

**Data Availability Statement:** The data utilized in this study may be acquired on demand from the corresponding author.

**Conflicts of Interest:** The authors declare no conflict of interest.

## Abbreviations

| | |
|---|---|
| *ALT* | Accelerated life testing |
| *BX* | Time which is an accumulated failure rate of X% |
| *CDF* | Cumulative distribution function |
| *D* | Driving force |
| *F* | Unreliability |
| *J* | Diffusion flux |
| *L* | Transport quantity |
| *LS* | Life-stress (LS) model |
| *TF* | Time to failure |
| *MTTF* | Mean time to failure |
| *R* | Reliability |

## Nomenclature

| | |
|---|---|
| $E_a$ | Activation energy, eV |
| $e$ | Effort |
| $e_b$ | Counterelectromotive force |
| $e_f$ | Field voltage, V |
| $f$ | Flow |
| $F_c$ | Ice-crushing force, kN |
| $F(t)$ | Unreliability |
| $h$ | Testing cycle |
| $h^*$ | Nondimensional testing time, $h^* = h/L_B \geq 1$ |
| $i_f$ | Field current, $A$ |
| $J$ | Momentum of inertia, kg m$^2$ |

| | |
|---|---|
| $k$ | Boltzmann's quantity, $8.62 \times 10^{-5}$ eV/deg |
| $L_B$ | Objective BX lifetime and $x = 0.01\ X$, on the condition that $x \leq 0.2$ |
| $m$ | Gear proportion |
| $MGY$ | Gyrator in causal forms |
| $n$ | Sample number |
| $Q$ | Entire number of dopants per unit area |
| $R$ | Ratio for minimum stress to greatest stress in stress cycle |
| $r$ | Failed numbers |
| $r$ | Coefficient of gyrator |
| $S$ | Stress |
| $T$ | Temperature, K |
| $ti$ | Test time for each sample |
| $T_L$ | Ice-crushing torque in bucket, kN cm |
| $X$ | Accumulated failure rate, % |
| $x$ | $x = 0.01\ X$, on condition that $x \leq 0.2$ |
| Greek symbols | |
| $\xi$ | Electrical field exerted |
| $\eta$ | Characteristic life |
| $\lambda$ | Cumulative damage quantity in Palmgren–Miner's rule |
| $\chi^2$ | Chi-square distribution |
| $\alpha$ | Confidence level |
| $\omega$ | Angular velocity in ice bucket, rad/s |
| Superscripts | |
| $\beta$ | Shape parameter on the Weibull chart |
| $n$ | Stress dependence, $n = -\left[\frac{\partial ln(T_f)}{\partial ln(S)}\right]_T$ |
| Subscripts | |
| 0 | Normal stress circumstances |
| 1 | Elevated stress circumstances |

## Appendix A. Derivation of Sample Size

To achieve the desired assignment cycles of ALT from the targeted BX lifetime in the testing scheme, the sample size formulation integrated with AF in Section 2.3 might be derived.

Each testing time, the Bernoulli test has one of the pair yields, such as failure or success. The accumulative probability, which keeps to a binomial distribution, is defined as follows:

$$L(p) = \sum_{r=0}^{c} \binom{n}{r} p^r \cdot (1-p)^{n-r} \leq \alpha \tag{A1}$$

where $n$ is the sample amount and $c$ is the presumed unsuccessful amount.

If chance $p$ is minute and $n$ is large enough, Equation (A1), which pursues a Poisson distribution, will be redefined:

$$L(n \cdot p) = \sum_{r=0}^{c} \frac{1}{r!}(n \cdot p)^r \cdot e^{-(np)} = \sum_{r=0}^{c} \frac{1}{r!} m^r \cdot e^{-m} \leq \alpha \tag{A2}$$

where $m = \text{parameter} = n \cdot p$.

As the $p$ amount is $\alpha$ from Equation (A2), parameter $m$ pursues the chi-square distribution, $\chi_\alpha^2()$. That is,

$$m = n \cdot p \sim \frac{\chi_\alpha^2(2r+2)}{2} \tag{A3}$$

The Weibull distribution for system lifetime is extensively employed because it is defined as an expression of the characteristic life, $\eta$, and shape parameter, $\beta$. Therefore, if the system keeps to the Weibull distribution, the accumulative failure rate, $F(t)$, in Equation (1) is defined as

$$F(t) = 1 - e^{-\left(\frac{t}{\eta}\right)^\beta} \tag{A4}$$

where $t$ is the (passed) time.

In the event of unreliability, $p = F(t)$, and reliability, $1 - p = R(t)$, Equation (A4) shall be placed into Equation (A1). That is,

$$L(p) = \sum_{r=0}^{c} \binom{n}{r} \left(1 - e^{-(\frac{t}{\eta})^{\beta}}\right)^{r} \cdot \left(e^{-(\frac{t}{\eta})^{\beta}}\right)^{n-r} \leq \alpha \tag{A5}$$

Because $e^{-(\frac{t}{\eta})^{\beta}} \cong 1 - \left(\frac{t}{\eta}\right)^{\beta}$, Equation (A5) can be closed as follows:

$$L(p) \cong \sum_{r=0}^{c} \frac{1}{r!} \left(\frac{t}{\eta}\right)^{\beta r} \cdot \left(1 - \left(\frac{t}{\eta}\right)^{\beta}\right)^{n-r} \leq \alpha \tag{A6}$$

As Equations (A2) and (A6) have a close shape, the characteristic life with a confidence level of $100 (1 - \alpha)$ may be clarified:

$$m = n \cdot p = n \cdot \left(\frac{t}{\eta}\right)^{\beta} \sim \frac{\chi_{\alpha}^{2}(2r + 2)}{2} \text{ or } \eta_{\alpha}^{\beta} = \frac{2}{\chi_{\alpha}^{2}(2r + 2)} \cdot n \cdot t^{\beta} \tag{A7}$$

At BX life, $L_B$, in Equation (A4), test cycles, $t$, becomes $h$.

$$L_{B}^{\beta} \cong x \cdot \eta_{\alpha}^{\beta} = x \cdot \frac{2}{\chi_{\alpha}^{2}(2r + 2)} \cdot n \cdot t^{\beta} = x \cdot \frac{2}{\chi_{\alpha}^{2}(2r + 2)} \cdot n \cdot h^{\beta} \geq L_{B}^{*\beta} \text{ for } x \leq 0.2 \tag{A8}$$

where $x = 0.01F(t)$.

If Equation (A8) is reordered, the sample size expression is found as:

$$n \geq \frac{\chi_{\alpha}^{2}(2r + 2)}{2} \times \frac{1}{x} \times \left(\frac{L_{B}^{*}}{h}\right)^{\beta} \tag{A9}$$

As the 1st term $\frac{\chi_{\alpha}^{2}(2r+2)}{2}$ in a 60% confidence level is approximated to $(r + 1)$, Equation (A9) is redefined as:

$$n \geq (r + 1) \times \frac{1}{x} \times \left(\frac{L_{B}^{*}}{h}\right)^{\beta} \tag{A10}$$

## Appendix B. Derivation of Governing Equation

To attain the differential equations that are made up of state variables for the parametric prototype, the bond graph in Figure 8 shall be settled at each node:

$$df \times E_2/dt = 1/L_a \times eE_2 \tag{A11}$$

$$df M_2/dt = 1/J \times eM_2 \tag{A12}$$

where $L_a$ is the electromagnetic inductance.

The junction from Equation (A11) is

$$eE_2 = e_a - eE_3 \tag{A13}$$

$$eE_3 = R_a \times fE_3 \tag{A14}$$

where $e_a$ is the exerted voltage and $R_a$ is the (electromagnetic) resistance.

The junction from Equation (A12) is

$$eM_2 = eM_1 - eM_3 \tag{A15}$$

$$eM_1 = (K_a \times i) - T_{Pulse} \tag{A16}$$

$$eM_3 = B \times fM_3 \tag{A17}$$

where $B$ is the viscous friction constant, and $k_a$ is the constant of the counterelectromotive force.

Because $fM_1 = fM_2 = fM_3 = \omega$ and $i = fE_1 = fE_2 = fE_3 = i_a$ from Equations (A13) and (A14),

$$eE_2 = e_a - R_a \times fE_3 \tag{A18}$$

$$fE_2 = fE_3 = i_a \tag{A19}$$

If Equations (A18) and (A19) are substituted into Equation (A11), then

$$di_a/dt = 1/L_a \times (e_a - R_a \times i_a) \tag{A20}$$

From Equations (A15)–(A17), we can attain

$$eM_2 = [(K_a \times i) - T_L] - B \times fM_3 \tag{A21}$$

$$i = i_a \tag{A22}$$

$$fM_3 = fM_2 = \omega \tag{A23}$$

If Equations (A21)–(A23) are substituted into (A12), then

$$d\omega/dt = 1/J \times [(K_a \times i) - T_L] - B \times \omega \tag{A24}$$

From Equations (A20) and (A24), the state equations can be attained as follows:

$$\begin{bmatrix} di_a/dt \\ d\omega/dt \end{bmatrix} = \begin{bmatrix} -R_a/L_a & 0 \\ mk_a & -B/J \end{bmatrix} \begin{bmatrix} i_a \\ \omega \end{bmatrix} + \begin{bmatrix} 1/L_a \\ 0 \end{bmatrix} e_a + \begin{bmatrix} 1 \\ -1/J \end{bmatrix} T_L \tag{A25}$$

As the differential equation in Equation (A25) finds the integral, the output, $y_p$, harvested by the ice-maker is obtained as follows:

$$y_p = \begin{bmatrix} 0 & 1 \end{bmatrix} \begin{bmatrix} i_a \\ \omega \end{bmatrix} \tag{A26}$$

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
