# Peer review of "Enhancing the Fatigue Design of Mechanical Systems Such as Refrigerator to Reserve Food in Agroindustry for the Circular Economy"

_sustainability, doi:10.3390/su15087010_

Round 1
Reviewer 1 Report
This paper reports a (generalized) life-stress prototype and sample size for improving the fatigue design of mechanical products. The improvements in the reliability of a refrigerator ice-maker, comprising an auger motor with bearing, were utilized as a case investigation. The topic of this paper is interesting and the scope of this paper falls into the Journal of Sustainability. However, many comments should be detailed and clarified.
1. The introduction of the paper is too redundant, and the author is advised to simplify it.
2. The figures in the paper are not clear, and it is recommended that the quality of all figures be improved.
3. The font size of the pictures in the paper is not uniform, and it is suggested that the author standardize it.
4. The authors just present some general problems that have been widely studied during recent years. What is the main research contribution of the paper?
5. The theoretical formulas listed in this paper are too redundant, and it is suggested that the author simplify them.
6. The structure of the article is not clear enough, and the authors are advised to highlight the main work of the paper.
7.The novelty of this paper is not enough.
Author Response
Dear Sirs:
Thank you for your review of our paper. As using the “Track Changes” function” in word file, we modified the paper based on comments from the editor. Below is a response to your review.
Comment 1. The introduction of the paper is too redundant, and the author is advised to simplify it
Response: In the introduction and conclusion we added some explanation for the novelty (or usefulness) of our work in accordance the purpose of this special issue, considering the useful life of refrigerators and the importance of their use in agroindustry.
Comment 2. The figures in the paper are not clear, and it is recommended that the quality of all figures be improved.
Response: As you recommend, we modified the quality of Figures such as Figure 1, Figure 4, Figure 5, Figure 6, Figure 7, Figure 9, Figure 10, Figure 11, Figure 12, Figure 13, and Figure 14.
We eliminated Figure 2 because it is explained in the text and is not necessary.
Comment 3. The font size of the pictures in the paper is not uniform, and it is suggested that the author standardize it.
Response: As you recommend, we modified the font size to improve the quality of Figures.
Comment 4. The authors just present some general problems that have been widely studied during recent years. What is the main research contribution of the paper?
Response: To show the main research contribution of the paper, we add a logical diagram for parametric ALT (graphical abstract) in the appendix A. Please check it
Comment 5. The theoretical formulas listed in this paper are too redundant, and it is suggested that the author simplify them.
Response: Because the deduction from eq. J's Schrodinger is unnecessary, we eliminated the deduction portion from eq. J's Schrodinger and add some references.
Comment 6. The structure of the article is not clear enough, and the authors are advised to highlight the main work of the paper.
Response: As you recommend, we modified our manuscript as follows:
- Improve English. Dr. Dennis L. O’Neal as author and native speaker checked and modified the manuscript to improve English.
- Add the novelty of the work, highlighting the reason for its inclusion in the special issue. In the introduction and conclusion, we added some explanation for the novelty (or usefulness) of our work in accordance the purpose of this special issue. We also modified the title of paper as follows: “Improving the fatigue design of mechanical products such as refrigerator to reserve food in agroindustry for the circular economy”. Please check it.
- Add a logical diagram. We add a logical diagram for parametric ALT (graphical abstract) in the appendix A.
- We separate a nomenclature and abbreviations section.
- To improve their quality of figures, we modified the quality of Figures such as Figure 1, Figure 4, Figure 5, Figure 6, Figure 7, Figure 9, Figure 10, Figure 11, Figure 12, Figure 13, and Figure 14. Figure 2 is eliminated.
- We eliminated the deduction portion from eq. J's Schrodinger and add some references.
- We separated the summary and conclusions
Comment 7. The novelty of this paper is not enough.
Response: As previously mentioned, In the introduction and conclusion we added some explanation for the novelty (or usefulness) of our work in accordance the purpose of this special issue, considering the useful life of refrigerators and the importance of their use in agroindustry.
We hope these address your concerns in your review.
Sincerely,
The authors
Reviewer 2 Report
The authors have written the paper/article by copying very large segments, including paragraphs, equations, figures, etc. of their paper(s) published before, i.e.:
Woo, S.; O’Neal, D.L.; Matvienko, Y.G.; Mebrahtu, G., Enhancing the fatigue of mechanical systems such as dispensers entrenched on generalized life-stress models and sample sizes, Appl. Sci. 2023, 13, 1358
Woo, S.; O’Neal, D.L.; Hassen, Y.M., Systematic methods to increase the lifetime of mechanical products such as refrigerators by employing parametric accelerated life testing, Appl. Sci. 2022, 12, 7484.
Certain differences include variations of expressions, or terms used, i.e., prolong or lengthen, estimate or approximate, verify or prove, prototype or model, advise or suggest, fulfilled or attained, understanding or comprehending, issues or problems, grow or enhance, etc. This has resulted in the Introduction part of the paper remaining almost the same, and in the end has implicated the researched cited literature to remain mostly unchanged (over 90 %).
Although research in the paper is concentrated on particular mechanical parts as the case studies of refrigerator components: motor bearings, in contrast to the case-studies in previous articles: ice-maker motor gear parts, or water dispenser, the approaches, methods, procedures used are the same. Apparently, at least over 40 (of 44) equations are the same, at least about 12 (of 15) figures are the same, and finally, 3 (out of 4) stated conclusions are the same.
I am not an expert on the publication ethics and antiplagiarism proof/verification methods applied in evaluating paper/article originality. I do understand that particular case-studies, originating from the same unit or component, involve the very same methodologies, equations, diagrams, etc. I find that the authors have included research that differs from the previously published research to an extent that allows the paper to be accepted. As publication ethics are concerned, they are specific to every particular journal policy. Since the repeating of the paper structure, paragraphs, equations, figures, and conclusions is prevalent, I leave the decision on this matter to the members of the journal Editorial.
Author Response
Dear Sirs:
Thank you for your review of our paper. As using the “Track Changes” function” in word file, we modified the paper based on comments from the editor. Below is a response to your review.
Comment 1. The authors have written the paper/article by copying very large segments, including paragraphs, equations, figures, etc. of their paper(s) published before, i.e.:
Woo, S.; O’Neal, D.L.; Matvienko, Y.G.; Mebrahtu, G., Enhancing the fatigue of mechanical systems such as dispensers entrenched on generalized life-stress models and sample sizes, Appl. Sci. 2023, 13, 1358
Woo, S.; O’Neal, D.L.; Hassen, Y.M., Systematic methods to increase the lifetime of mechanical products such as refrigerators by employing parametric accelerated life testing, Appl. Sci. 2022, 12, 7484.
Certain differences include variations of expressions, or terms used, i.e., prolong or lengthen, estimate or approximate, verify or prove, prototype or model, advise or suggest, fulfilled or attained, understanding or comprehending, issues or problems, grow or enhance, etc. This has resulted in the Introduction part of the paper remaining almost the same, and in the end has implicated the researched cited literature to remain mostly unchanged (over 90 %).
Although research in the paper is concentrated on particular mechanical parts as the case studies of refrigerator components: motor bearings, in contrast to the case-studies in previous articles: ice-maker motor gear parts, or water dispenser, the approaches, methods, procedures used are the same. Apparently, at least over 40 (of 44) equations are the same, at least about 12 (of 15) figures are the same, and finally, 3 (out of 4) stated conclusions are the same.
I am not an expert on the publication ethics and antiplagiarism proof/verification methods applied in evaluating paper/article originality. I do understand that particular case-studies, originating from the same unit or component, involve the very same methodologies, equations, diagrams, etc. I find that the authors have included research that differs from the previously published research to an extent that allows the paper to be accepted. As publication ethics are concerned, they are specific to every particular journal policy. Since the repeating of the paper structure, paragraphs, equations, figures, and conclusions is prevalent, I leave the decision on this matter to the members of the journal Editorial.
Response: Because submitting our manuscript, we modified lots of sentences, equations, etc. from our previous manuscript.
To stress the novelty of the work, highlighting the reason for its inclusion in the special issue (consider the useful life of refrigerators and the importance of their use in agroindustry), we modified our manuscript as follows:
- Improve English. Dr. Dennis L. O’Neal as author and native speaker checked and modified the manuscript to improve English.
- Add the novelty of the work, highlighting the reason for its inclusion in the special issue. In the introduction and conclusion, we added some explanation for the novelty (or usefulness) of our work in accordance the purpose of this special issue. We also modified the title of paper as follows: “Improving the fatigue design of mechanical products such as refrigerator to reserve food in agroindustry for the circular economy”. Please check it.
- Add a logical diagram. We add a logical diagram for parametric ALT (graphical abstract) in the appendix A.
- We separate a nomenclature and abbreviations section.
- To improve their quality of figures, we modified the quality of Figures such as Figure 1, Figure 4, Figure 5, Figure 6, Figure 7, Figure 9, Figure 10, Figure 11, Figure 12, Figure 13, and Figure 14. Figure 2 is eliminated.
- We eliminated the deduction portion from eq. J's Schrodinger and add some references.
- We separated the summary and conclusions
We hope these address your concerns in your review.
Sincerely,
The authors

Reviewer 3 Report
Dear Editor
Current work describes the improving the fatigue design of mechanical products such as bearings based on a (generalized) life-stress prototype and sample size. I think it is out of scope of the sustainability journal and has a good potential for consideration in the materials journal.
Author Response
Dear Sirs:
Thank you for your review of our paper. As using the “Track Changes” function” in word file, we modified the paper based on comments from the editor. Below is a response to your review.
Comment 1. Current work describes the improving the fatigue design of mechanical products such as bearings based on a (generalized) life-stress prototype and sample size. I think it is out of scope of the sustainability journal and has a good potential for consideration in the materials journal.
Response: To stress the novelty of the work, highlighting the reason for its inclusion in the special issue (consider the useful life of refrigerators and the importance of their use in agroindustry).
We modified our manuscript as follows:
- Improve English. Dr. Dennis L. O’Neal as author and native speaker checked and modified the manuscript to improve English.
- Add the novelty of the work, highlighting the reason for its inclusion in the special issue. In the introduction and conclusion, we added some explanation for the novelty (or usefulness) of our work in accordance the purpose of this special issue. We also modified the title of paper as follows: “Improving the fatigue design of mechanical products such as refrigerator to reserve food in agroindustry for the circular economy”. Please check it.
- Add a logical diagram. We add a logical diagram for parametric ALT (graphical abstract) in the appendix A.
- We separate a nomenclature and abbreviations section.
- To improve their quality of figures, we modified the quality of Figures such as Figure 1, Figure 4, Figure 5, Figure 6, Figure 7, Figure 9, Figure 10, Figure 11, Figure 12, Figure 13, and Figure 14. Figure 2 is eliminated.
- We eliminated the deduction portion from eq. J's Schrodinger and add some references.
- We separated the summary and conclusions
We hope these address your concerns in your review.
Sincerely,
The authors
Round 2
Reviewer 1 Report
The authors have revised the paper and addressed the reviewer's comments effectively.
Author Response
-
Reviewer 2 Report
Although the authors have made certain corrections to the paper/article, the comments I have included were based on publication ethics and antiplagiarism proof/verification, allegedly, the paper still being remarkably similar to the earlier two published papers, as stated before. The authors did not react to my comments concerning the large portion of the paper having equations; figures; stated conclusions, that are repeated to the extent that brings on judgement over publication ethics.
Author Response
Dear Sirs:
Thank you for your review of our paper. As using the “Track Changes” function” in word file, we modified the paper based on comments from the reviwer. Below is a response to your review.
Comment 1. Although the authors have made certain corrections to the paper/article, the comments I have included were based on publication ethics and antiplagiarism proof/verification, allegedly, the paper still being remarkably similar to the earlier two published papers, as stated before. The authors did not react to my comments concerning the large portion of the paper having equations; figures; stated conclusions, that are repeated to the extent that brings on judgement over publication ethics
Response: As you raised some doubts, we modified a lot of changes to equations and diagrams as follows:
Equations: Equation (2), Equation (3), Equation (5), Equation (6), Equation (8), Equations (10)-(13), Equations (19)-(30), Equations (31)-(34),
That is, after rederiving the sample size equation in section 2.4, we moved the original sample size equations to appendix B. We also moved the derivation of the differential equations of ice-maker to appendix C.
Diagrams: Figure 1, Figure 2, Figure 5, Figure 8. Figure 10, and Figure 13.
And we altered the conclusions.
We hope these address your concerns in your review.
Sincerely,
The authors
Reviewer 3 Report
It can be published in current form.
Author Response
-
Round 3
Reviewer 2 Report
There are no further comments or recommendations to the authors. The authors have made numerous changes to the manuscript, that may qualify it for publication.